# Gender-Specific Associations between Late-Life Disability and Socioeconomic Status: Findings from the International Mobility and Aging Study (IMIAS)

**DOI:** 10.3390/ijerph20042789

**Published:** 2023-02-04

**Authors:** Luana Caroline de Assunção Cortez Corrêa, Cristiano dos Santos Gomes, Saionara Maria Aires da Camara, Juliana Fernandes de Souza Barbosa, Ingrid Guerra Azevedo, Afshin Vafaei, Ricardo Oliveira Guerra

**Affiliations:** 1Department of Physical Therapy, Federal University of Rio Grande do Norte, Natal 59078-140, Brazil; 2Department of Physical Therapy, Federal University of Pernambuco, Recife 50670-901, Brazil; 3Departamento de Procesos Terapéuticos, Facultad de Ciencias de la Salud, Universidad Catolica de Temuco, Temuco 4813302, Chile; 4School of Health Studies, Western University, London, ON N6A 3K7, Canada

**Keywords:** disability, socioeconomic status, older adults, gender, population-based study

## Abstract

Disability is a dynamic process and can be influenced by a sociocultural environment. This study aimed to determine whether the associations between socioeconomic status and late-life disability differ by gender in a multi-sociocultural sample from different countries. A cross-sectional study was developed with 1362 older adults from The International Mobility in Aging Study. Late-life disability was measured through the disability component of the Late-Life Function Disability Instrument. Level of education, income sufficiency and lifelong occupation were used as indicators of SES. The results indicated that a low education level β = −3.11 [95% CI −4.70; −1.53] and manual occupation β = −1.79 [95% −3.40; −0.18] were associated with frequency decrease for men, while insufficient income β = −3.55 [95% CI −5.57; −1.52] and manual occupation β = −2.25 [95% CI −3.89; −0.61] played a negative role in frequency for women. For both men β = −2.39 [95% −4.68; −0.10] and women β = −3.39 [95% −5.77; −1.02], insufficient income was the only factor associated with greater perceived limitation during life tasks. This study suggested that men and women had different late-life disability experiences. For men, occupation and education were associated with a decrease in the frequency of participation, while for women this was associated with income and occupation. Income was associated with perceived limitation during daily life tasks for both genders.

## 1. Introduction

Disability is defined as an impairment, inability or limitation in social performance tasks considering the sociocultural environment in which one lives [1]. As a dynamic process, disability can be influenced by various factors, such as a sedentary lifestyle, elevated body mass index and poor nutrition, among others [2,3]. Moreover, ample attention has been given to how discrepancies in socio-economic status (SES), including income, education and occupation, affect disability in late life [4,5]. Functional decline and disability processes increase significantly with advanced age [1] and are a great burden on health systems around the world [6].

Despite epidemiological studies with large representative samples showing that it has declined over the past few decades [7,8], disability remains highly prevalent [1]. According to data from 54 countries assessed in the World Health Survey (WHS), disability prevalence stands at 39% among older persons (aged 65 or over), with women showing higher prevalence than men [9]. The difference in prevalence rates between genders could be explained by sex-linked factors that render women more susceptible to adverse health outcomes at advanced ages, such as reproductive history (multiparty, early maternal age at first birth and menopause) [10,11]. However, evidence suggests that, beyond biological differences, the social structure of a culture that includes the division of gender-related roles, social functions and social status has been a more important factor in determining gender differences in late-life disability [12,13].

Ample attention has been given to how socioeconomic discrepancies affect disability [14]. Previous findings have shown that disability differs between high-income and low/middle-income countries and urban and rural areas, since better socioeconomic conditions, income and environment can influence adequate access to health and social care, as well as community participation [14,15]. Furthermore, developed countries tend to advance more in terms of gender equality, which can contribute to reducing culturally related factors and social barriers that affect women’s health more strongly than men [16]. In this sense, disability results from an interaction between individual and environmental aspects, mediated by the social and cultural context [17].

To develop and support prevention strategies, there is a consensus on urgently evaluating the disability process in the older population in different settings. However, comprehensive tools that capture a diverse range of function activities and the process of disability change are necessary. The Late Life Function and Disability Instrument (LLFDI) was developed as a self-report instrument for use by older adults living in the community [18], and to assess function and disability constructs based on the WHO International Classification of Functioning, Disability and Health (ICF), mainly for participation-level measures [19]. The LLFDI has good reproducibility and validation in multiple populations, without ceiling and floor effects to assess disability in older populations [18,20]. Furthermore, it has demonstrated concurrent and predictive validity with functionally objective assessment measures [21].

Although previous studies have found late-life disability to be associated with socioeconomic status [7,22] and that there are differences in late-life disability both by gender [23,24] and by socioeconomic status [12], little is known about whether the association between late-life disability and socioeconomic status differs by gender. Thus, we aimed to assess the associations between late-life disability and socioeconomic status from a gender perspective in a sample of community-dwelling older adults from three countries (Canada, Brazil and Colombia). The IMIAS multicenter cohort study [25] is a great resource for analyzing associations between gender, SES and late-life disability, since the sample represents a diverse range of lifestyles in different sociocultural and economic contexts. Our objective is to discover whether the associations between socioeconomic status and late-life disability differ by gender.

## 2. Materials and Methods

The International Mobility in Aging Study (IMIAS) is a longitudinal multi-site study aiming to evaluate the effects of gender differences on the mobility of community-dwelling older adults in international cities that differ in culture, economy, physical environment and healthcare system. Additional details about the study sites and procedures are available in a previous publication [25] and on the website: www.imias.ufrn.br (accessed on 20 October 2022). 

### 2.1. Sampling Strategy and Data Collection

The population samples in Natal and Manizales were randomly selected through registries at community health clinics and public health insurance databases. Due to contact restriction imposed by ethics committees in Kingston and Saint-Hyacinthe, potential participants from those cities were invited by letter from their primary care physicians to contact our field coordinator in order to participate in the study. In 2012, a sample of 1608 older people living in the community, 65 to 74 years old (approximately 200 men and 200 women), were recruited from Natal (Brazil), Manizales (Colombia), Kingston (Ontario, Canada) and Saint-Hyacinthe (Quebec, Canada). The first follow-up assessment was conducted in 2014 and the second in 2016. For this study, we used data from 1362 participants evaluated in 2014, since information about late-life disability was collected in this wave.

### 2.2. Ethical Considerations

This study was approved by local ethics boards at the respective sites, and all participants were informed about research procedures and gave their informed consent in writing.

### 2.3. Measures

#### 2.3.1. Late-Life Disability

The primary outcome was assessed by the disability component of the LLFDI, which measures self-reported frequency (objective) of participation and perceived limitation (subjective) in 16 life-task situations [26]. Both subscales (frequency and limitation) have demonstrated current and predictive validity with functional performance measures among community-dwelling older adults [18,26,27], have good reproducibility validation in multiple populations [20] and have been validated using IMIAS data [21].

The frequency subscale dimension refers to the individual’s regularity in participating in social and personal activities. It is assessed by the question, “How often do you do a particular task?”, with the following rating scale categories: “Very often (5 points)”, “Often (4 points)”, “Once in a while (3 points)”, “Almost never (2 points)” and “Never (1 point)”.

The limitation subscale dimension refers to the perceived limitation of the individual in performing instrumental and management activities, considering internal and external factors that may influence them, such as health, physical and mental energy, transportation, accessibility and socioeconomic condition. It is assessed using the question, “To what extent do you feel limited in doing a particular task?”, with the following rating scale categories: “Not at all (5 points)”, “A little (4 points)”, “Somewhat (3 points)”, “A lot (2 points)” and “Completely (1 point)”.

The LLFDI score is calculated in two stages. First, raw summary scores of frequencies and limitations are found, ranging from 16 to 80 points (all 16 items). Second, raw summary scores are transformed to scaled summary scores, ranging from 0 to 100 points. Scores approaching 100 signify high levels of frequency and ability to participate in life tasks and less limitation [26].

#### 2.3.2. Socioeconomic Status (SES)

We used the three most robust indicators of SES: income, education and occupation [28]. Perceived income sufficiency was used as a measure of income by asking, “To what extent does your income allow you to meet your needs?” The responses were coded as “not/not very well”, “suitably” and “very well”. We adopted this measure instead of current income to characterize SES because we believed that, in many cases, such as when there are multiple dependents, the participant’s income may not reflect their ability to meet their needs [29]. The level of education was collected as years of study and coded into three categories—“less than secondary”, “secondary” and “post-secondary”—according to the 25th and 75th percentiles of each research site. This categorization was necessary for cross-site comparisons due to the ample differences in years of schooling among the sites of study [29]. Lifelong occupation was evaluated with open questions and classified according to the International Standard Classification of Occupations, in 10 major groups, then further classified into two groups: “non manual” and “manual” [30].

#### 2.3.3. Covariates

Analytical models were adjusted for participant age, number of chronic diseases, self-rated health and depression symptoms. Age was recorded in years. Information on self-reported general health status was collected using the following question: “Would you say that your health is: very good, good, fair, poor, very poor?” The answers were then categorized into two groups: “very good/good” and “fair/poor/very poor”. Self-rated health (SRH) is a strong and valid indicator of overall health [31]. Depression was assessed using the Center for Epidemiology Studies Depression Scale (CES-D), which consists of a 20-item scale measuring depressive symptoms experienced over the past week [32]. The CES-D has been widely used as a valid measure for older populations in different settings, and the standard cut-off score of 16 or more is used to suggest depressive symptoms [33]. Information about chronic conditions was collected using the question, “Has a doctor or nurse ever told you that you have a chronic disease?” The question refers to hypertension, diabetes, cancer, chronic lung disease, heart disease, stroke (cerebral embolism or thrombosis), osteoarthritis (arthritis or rheumatism), osteoporosis and cataract. Responses were classified into three groups: “one or fewer”, “two or three” and “four or more conditions”.

### 2.4. Statistical Analysis

All data were analyzed using the IBM SPSS statistical software program version 20.0. First, the characteristics of the sample were compared by gender and described using means (standard deviations, SD) for continuous variables and frequencies (percentages) for categorical variables. The chi-square test (χ^2^) and Student’s were used to compare categorical variables and continuous variables between groups, respectively. Second, Student’s *t* test and analysis of variance (ANOVA) were used to compare the means of frequency and limitation between men and women with each independent variable. Finally, multiple linear regression models were used to examine gender-specific associations between late-life disability (frequency and limitation subscales) and SES. We adjusted multiple linear regression models for the following variables based on precedents suggesting that they were risk factors for late-life disability while also being related to SES measures: age, number of chronic diseases, self-rated health and depressive symptoms [23,34,35]. A 5% level of statistical significance was used for all analyses and estimates were reported with their 95% confidence intervals (CI).

## 3. Results

### 3.1. Sample Characteristics

The sample consisted of 1362 participants (721 women) and the mean ages for men and women were similar (71.13 SD 2.94 vs. 71.27 SD 2.80, *p* = 0.375). Marital status varied significantly by gender, with more men than women married (81.4% vs. 48.7%, *p* < 0.001). More women than men reported four or more chronic diseases (13.8% vs. 5.8%, *p* < 0.001), poor self-rated health (36.6% vs. 29.7%, *p* = 0.001) and depressive symptoms (17.3% vs. 8.1%, *p* = 0.007) (Table 1). The gender differences between SES were statistically significant only in educational level, with men having a higher level of education than women (36.7% vs. 23.1%, *p* < 0.001). Approximately similar proportions of men and women worked in manual occupations (59.9% vs. 59.5%, *p* = 0.877) and perceived their income’s ability to meet their needs as not/not very well (34.4% vs. 37.2% *p* = 0.452). More details of the characteristics of the participants are shown in Table 1.

### 3.2. Late-Life Disability, SES, and Gender

Late-life disability varied significantly by gender (Table 2). The mean summary scores of the frequency subscale were significantly lower for men compared to women (53.95 SD 9.05 vs. 56.74 SD 9.31, *p* < 0.001). On the other hand, the mean summary scores of the limitation subscale were lower among women compared to men (72.95 SD 9.86 vs. 75.20 SD 8.74, *p* < 0.001). Overall, among research sites, men had disadvantages in terms of frequency in life tasks compared to women, except in St. Hyacinthe (*p* = 0.273). In both men and women, the mean summary scores on the frequency subscale were lower among those who were less educated, held a manual occupation, had insufficient income, had a higher number of chronic diseases, had fair/very poor self-rated health and reported depressive symptoms. Regarding the limitation subscale, among all research sites, women reported more limitations in their ability to perform life tasks compared to men, being statistically significant in Natal (*p* = 0.006) and St. Hyacinthe (*p* = 0.002). In both men and women, the severity of limitation was greater among those who held a manual occupation, reported insufficient income, had four or more chronic diseases, had fair poor/very poor self-rated health and reported depressive symptoms.

### 3.3. Late-Life Disability, SES, and Gender—Frequency and Limitation Subscales

The results of the multiple regression analyses are presented in Table 3 and Table 4. In the fully adjusted model (Model 3) the mean summary scores of the frequency subscale decreased for all research sites compared to Kingston for both genders. Low education level β = −3.11 [95% CI −4.70; −1.53] and manual occupation β = −1.79 [95% −3.40; −0.18] are factors associated with a decrease in frequency for men, while insufficient income β = −3.55 [95% CI −5.57; −1.52] and manual occupation β = −2.25 [95% CI −3.89; −0.61] played a negative role in frequency for women (Table 3). Surprisingly, the mean summary scores of the limitation subscale increased for all research sites compared to Kingston for men and women, especially among those living in Natal β = 4.28 [95% CI 1.74; 6.81] and Manizales β = 6.35 [95% CI 3.96; 8.74]; Natal β = 4.94 [95% CI 2.12; 7.77] and Manizales β = 8.12 [95% CI 5.38; 10.86], respectively. For both men β = −2.39 [95% −4.68; −0.10] and women β = −3.39 [95% −5.77; −1.02], insufficient income was the only factor associated with increased perceived limitation during life tasks (Table 4).

## 4. Discussion

The present study investigated the association between late-life disability and SES by gender using a representative sample of IMIAS. Our findings indicated that men and women had different late-life disability experiences, with men reporting greater community disability and women reporting greater perceived disability. Moreover, our results highlighted a gender-specific association between the indicators of SES, including education, income and occupation, and late-life disability.

Analyzing the frequency subscale reveals that men had lower scores compared to women, indicating less participation in life tasks and therefore more disability. Although some population studies evaluating gender differences in disability had found similar findings, it is documented that men have higher rates of incidence and prevalence of cardiovascular diseases [36], especially stroke and heart attack, which may lead to physical and cognitive consequences contributing to disability. Moreover, men are more likely to adopt unhealthy behaviors during the course of their life [37] such as smoking, alcohol consumption and drug abuse, which have been shown to be associated with an increased risk of physical disability [38]. However, it is important to note that, in our multicultural sample with participants from low/middle- and high-income countries, which presents diverse sets in terms of gender equality [39,40], this difference may exist due to the nature of the questionnaire content, which mainly includes tasks, responsibilities and attributes related to socially constructed gender roles, resulting in negative responses among men that may not reflect the real presence of disability [26]. For instance, many older men habitually do not prepare meals, perform household tasks, visit friends and family in their homes, provide care or assistance to others, etc., instead leaving these activities to their wives [41].

On the other hand, women had lower scores in the limitation subscale compared to men, indicating less self-perceived capability in performing life tasks and therefore more disability. Our finding is consistent with large representative studies that have extensively reported a higher prevalence of disability among women, measured by limitations in functional upper and lower extremities and difficulties in activities of daily living (basic and instrumental) [23,24,42]. Part of the explanation for this may be that women in later life have a higher prevalence of comorbidity, depression and cognitive impairment, which may lead to an increased perception of limitations and restrictions in activities [43]. It can also be understood from a biological perspective involving hormones, body composition, nutritional problems (obesity, undernutrition) and loss of muscle mass (sarcopenia), all of which affect older women more than men [11,43]. Furthermore, women are more frequently exposed to adverse conditions throughout their lives, such as physical and psychological violence, discrimination and inequality of opportunity compared to men [44].

Continuous exposure to the aforementioned factors can create attitudinal barriers and increase the perception of limitation among women [45]. Corroborating these findings, Sayers et al., 2004, examined the concurrent and predictive validity of the disability component of LLDFI in a sample of older adults and demonstrated that only the limitation subscale demonstrated predictive validity with functional performance measures assessed by the short physical performance battery, 400-m walk test and gait speed [18].

This study showed that the impact of SES on late-life disability is heterogeneous among gender. For men, the association between SES and the frequency of participation in life tasks was worsened by having low education levels and having had a manual occupation. These are two important social dimensions of health inequalities and have been associated with adverse outcomes in older age, such as functional disability, cognitive decline, low quality of life and impaired social functioning [46,47]. There are many possible explanations for why occupation and education might influence late life disability. Manual occupation is generally associated with poor health status due to the increased exposure of individuals to physical and environmental hazards such as noise, heat, cold, vibration, high physical demands, repetitive work and psychosocial stressors [47]. Furthermore, manual occupation is likely to be related to the low material resources that, in the long term, can have an impact on health resulting in disability [48]. Education, on the other hand, is one of the most important factors that has contributed to changes in late-life disability trends in recent decades, and a change in education is positively associated with a reduction in the onset of disability in activities of daily living (ADL) [22]. It is possible that greater educational attainment can improve healthy behaviors, health care quality and greater use of assistive technologies [49]. In addition, less educated people may have precarious access to services and programs which could improve their participation in community life and their engagement in physical and recreational activities [49].

Regarding women, the results of this study indicate that frequency of participation in life tasks decreased significantly among those who reported manual occupation and insufficient income. Our findings support a growing body of literature that shows that the poorest women face higher disability rates [24,48,50] and highlights the role of financial resources as a barrier to involvement in social and personal activities [22]. Income has been shown to be a key factor in the onset and progression of disability [48].

Data from the 2001–2014 Canadian Community Health Survey (CCHS) showed that improving household income can reduce the prevalence rates of ADL disability, suggesting that high financial resources play a preventive role in disability and also decrease the accumulation of disability over time, since an individual can improve access to material resources such as assistive technology, a wheelchair or hearing aids, among others [7]. Ashida et al. (2016) similarly suggest that higher incomes are also associated with increased participation in volunteer groups, sports, hobbies and social networks, which can reduce the risk of disability [14]. The way in which manual occupation affects women’s health can differ compared to men, especially among those born in the first half of the twentieth century. While men are often engaged in physical and stressful jobs, women are more likely to be concentrated in some types of work that require repetitive movements [47,51] (i.e., industries and household workload) and are more often exposed to psychological violence (i.e., verbal aggression and moral harassment) and sexual abuse in their occupations [52]. Those circumstances have been previously associated with mental health and functioning outcomes [53] that may result in a lack of participation in regular activities, mobility disability and a decrease in social interactions.

After adjusting for all relevant confounding factors (age, number of chronic diseases, self-rated health and depressive symptoms), income insufficiency was the only SES independently associated with a greater perceived limitation in the ability to perform life tasks for both genders. It is important to emphasize that these variables (low education, manual work, dissatisfaction with income) are related throughout the course of one’s life, but under a cross-sectional analysis what will actually make the difference is the lack of income, because that is what they are experiencing at the time. Perhaps due to the survival/resilience effects of the studied samples and because of the age range of the study subjects (65–74 years), disabilities tend to appear in more advanced stages of old age.

Finally, we conceptualized the health status variables of a number of chronic diseases, self-rated health and depressive symptoms as potential confounders and therefore adjusted for them via multivariate models; however, the possibility of mediator roles cannot be ruled out. We repeated the analysis for both the frequency and limitation subscales with the exclusion of these variables, and minimal changes in the estimated coefficients were observed (Frequency subscale – Men: β = 53.82 [95% CI 39.70; 67.95); Women: β = 83.05 [95% CI 67.97; 98.12) (Limitation subscale – Men: β = 75.60 [95% CI 59.64; 91.56); Women: β = 94.11 [95% CI 75.84; 112.98). We decided to report the models that included these variables in the final results.

## 5. Strengths and Limitations

The strengths of our study include a large community sample of older adults, which allows comparison by gender. Furthermore, this study used a standardized protocol and measurement tools to assess how SES and late-life disability are associated with gender. The comprehension of disability used here covers a range of domains that are not found in many traditional measures, including participation in social and personal activities. However, our findings must be interpreted with caution. First, the cross-sectional nature of this study hinders causal inference, and it is only possible to extrapolate associations between SES and disability. Longitudinal studies and mediation analysis are particularly needed to understand how the relationship changes over time and the possibility of health variables (age, chronic diseases, depressive symptoms, and self-rated health) to act as mediators between SES and disability. Second, we used self-report measures, which may have resulted in some degree of reporting bias; nonetheless, strong evidence shows that they are reliable predictors and an important measure of health outcomes [54].

## 6. Conclusions

Men and women had different late-life disability experiences, with men reporting greater community disability and women reporting greater perceived disability. For men, occupation and education were associated with a decrease in frequency of participation, while for women it was associated with income and occupation. Income was associated with perceived limitation during daily life tasks for both genders. These findings are important to consider when formulating public policies and can be used to guide prevention and early interventions for vulnerable groups to reduce the impacts of late-life disability and promote social participation.

## Figures and Tables

**Table 1 ijerph-20-02789-t001:** Distribution of study survey participants according to socioeconomic and health characteristics (n = 1362).

Variables	Men (n = 641)	Women (n = 721)	*p*-Value
Age, M (SD)	71.13 (± 2.94)	71.27 (± 2.80)	*p* = 0.375
Study site, n (%)			*p* = 0.627
Kingston	146 (22.8%)	182 (25.2%)	
St. Hyacinthe	161 (25.1%)	184 (25.5%)	
Manizales	184 (28.7%)	188 (26.1%)	
Natal	150 (23.4%)	167 (23.2%)	
Marital status, n (%)			*p* < 0.001
Single	35 (5.5%)	68 (9.4%)	
Married	522 (81.4%)	351 (48.7%)	
Widow/divorced	84 (13.1%)	302 (41.9%)	
Education level, n (%) ^a^			*p* < 0.001
Less than secondary	220 (34.4%)	267 (37.0%)	
Secondary	185 (28.9%)	288 (39.9%)	
Post-secondary	235 (36.7%)	166 (23.1%)	
Occupation, n (%) ^b^			*p* = 0.877
Manual	384 (59.9%)	423 (59.5%)	
Non manual	257 (40.1%)	288 (40.5%)	
Income sufficiency, n (%) ^c^			*p* = 0.452
Not/not very well	220 (34.4%)	268 (37.2%)	
Suitably	218 (34.1%)	225 (31.3%)	
Very well	201 (31.5%)	227 (31.5%)	
Chronic diseases, n (%) ^d^			*p* < 0.001
≤1	355 (55.5%)	272 (37.8%)	
2–3	253 (39.5%)	349 (48.5%)	
≥4	32 (5.0%)	99 (13.8%)	
Self-rated health, n (%) ^e^			*p* = 0.007
Fair poor/very poor	190 (29.7%)	263 (36.6%)	
Very good/good	450 (70.3%)	456 (63.4%)	
Depressive symptoms, n (%)			*p* < 0.001
Depressed	52 (8.1%)	125 (17.3%)	
Not depressed	589 (91.9%)	596 (82.7%)	

^a^ One missing value; ^b^ Ten missing values; ^c^ Three missing values; ^d^ Two missing values; ^e^ Three missing values; M = Mean; SD = Standard deviation

**Table 2 ijerph-20-02789-t002:** Bivariate analysis between frequency and limitation subscales, socioeconomic and health characteristics by gender.

Variables	Frequency Subscale			Limitation Subscale		*p*-Value
	Men (n = 641)	Women (n = 721)	*p* < 0.001 *	Men (n = 641)	Women (n = 721)	
LLFDI disability component, M (SD)	53.95 (9.05)	56.74 (9.31)	*p* < 0.001 *	75.20 (8.74)	72.95 (9.86)	*p* < 0.001
Study sites, M (SD)						
Kingston	59.83 (6.16)	62.08 (7.31)	*p* = 0.003	72.95 (7.84)	71.54 (9.65)	*p* = 0.155
St. Hyacinthe	58.16 (7.17)	59.04 (7.63)	*p* = 0.273	76.34 (6.67)	73.74 (8.41)	*p* = 0.002
Manizales	50.33 (8.33)	54.99 (8.81)	*p* < 0.001	77.12 (9.37)	75.74 (8.14)	*p* = 0.130
Natal	48.19 (8.49)	50.36 (9.20)	*p* = 0.030	73.84 (8.92)	70.47 (12.23)	*p* = 0.006
Education level, M (SD)						
Less than secondary	51.08 (9.26)	54.61 (9.45)	*p* < 0.001	74.73 (9.05)	71.81 (10.99)	*p* < 0.002
Secondary	54.70 (9.13)	57.34 (9.24)	*p* = 0.002	74.83 (9.92)	73.30 (9.51)	*p* = 0.088
Post-secondary	56.09 (8.07)	59.08 (8.51)	*p* < 0.001	75.92 (6.43)	74.17 (8.27)	*p* = 0.018
Occupation, M (SD)						
Manual	50.76 (8.71)	53.62 (9.21)	*p* < 0.001	75.26 (9.05)	72.85 (10.86)	*p* = 0.001
Non manual	58.71 (7.70)	61.53 (7.14)	*p* < 0.003	75.12 (7.53)	73.10 (8.21)	*p* = 0.003
Income sufficiency, M (SD)						
Not/not very well	48.54 (7.94)	52.26 (9.32)	*p* < 0.001	74.76 (10.34)	71.85 (11.66)	*p* = 0.004
Suitably	54.58 (8.92)	56.72 (8.46)	*p* = 0.010	75.17 (8.54)	73.51 (9.21)	*p* = 0.050
Very well	59.24 (6.80)	62.03 (7.13)	*p* < 0.001	75.74 (6.38)	73.66 (7.91)	*p* = 0.002
Chronic diseases, M (SD)						
≤1	54.03 (8.60)	58.87 (8.76)	*p* < 0.001	76.27 (7.33)	75.37 (7.92)	*p* = 0.143
2–3	54.00 (9.75)	56.16 (9.00)	*p* = 0.005	74.33 (8.84)	72.83 (9.06)	*p* = 0.043
≥4	52.84 (8.53)	52.82 (10.32)	*p* = 0.990	71.00 (13.46)	66.78 (13.93)	*p* = 0.136
Self-rated health, M (SD)						
Fair poor/very poor	47.84 (8.85)	51.28 (9.25)	*p* < 0.001	71.57 (12.19)	69.14 (12.71)	*p* = 0.042
Very good/good	56.52 (7.83)	59.86 (7.18)	*p* < 0.001	76.75 (5.62)	75.11 (6.90)	*p* < 0.001
Depressive symptoms, M (SD)						
Depressed	47.60 (10.23)	51.41 (10.25)	*p* = 0.025	67.87 (14.96)	67.95 (13.08)	*p* = 0.970
Not depressed	54.52 (8.73)	57.84 (8.72)	*p* < 0.001	75.85 (7.31)	73.99 (8.70)	*p* < 0.001

* *p*-values comparing men and women at each independent variable; LLFDI = Late-Life Function and Disability Instrument; M = Mean; SD = Standard deviation.

**Table 3 ijerph-20-02789-t003:** Multivariate regression models assessing the influence of socioeconomic variables on the frequency subscale separated by gender.

	Men (n = 641)						Women (n = 721)					
	Model 1		Model 2		Model 3		Model 1		Model 2		Model 3	
	β	95% CI	β	95% CI	β	95% CI	β	95% CI	β	95% CI	β	95% CI
(Intercept)	59.79	[45.95;73.63]	54.63	[41.02;68.23]	54.10	[40.53;67.66]	88.54	[74.03;103.06]	86.51	[72.03;101.00]	85.24	[70.69;99.79]
Study site												
Natal	−7.57 ^c^	[−9.71,−5.43]	−8.25 ^c^	[−10.36;−6.14]	−7.23 ^c^	[−9.91;−5.70]	−5.46 ^c^	[−7.60;−3.33]	−6.23 ^c^	[−8.41;−4.06]	−4.62 ^c^	[−7.03;−2.21]
Manizales	−6.53 ^c^	[−8.61;−4.46]	−7.10 ^c^	[−9.14;−5.06]	−6.29 ^c^	[−8.44;−4.13]	−2.41 ^a^	[−4.46;−0.36]	−3.18 ^b^	[−5.27;−1.09]	−1.59	[−3.93;0.74]
St. Hyacinthe	−1.62	[−3.27;−0.02]	−1.37	[−2.98;0.23]	−0.90	[−2.56;0.76]	−2.39 ^b^	[−3.96;−0.73]	−2.44 ^b^	[−4.05;−.84]	−1.78 ^a^	[−3.45;−0.11]
Kingston	Ref.		Ref.		Ref.		Ref.		Ref.		Ref.	
Income sufficiency												
Not/not very well	−3.04^b^	[−5.10;−0.98]	−1.69	[−3.75;0.36]	1.40	[−3.47;0.67]	−4.48 ^c^	[−6.24;−2.72]	−3.64 ^c^	[−5.67;−1.62]	−3.55 ^b^	[−5.57;−1.52]
Suitably	−0.67	[−2.27;0.92]	0.17	[−1.40;1.75]	0.32	[−1.25;1.90]	−2.77 ^c^	[−4.25;−1.29]	−2.09 ^b^	[−3.68;-.50]	−1.83 ^a^	[−3.43;−0.23]
Very well	Ref.		Ref.		Ref.		Ref.		Ref.		Ref.	
Education level												
Less than secondary	-	-	−4.04 ^c^	[−5.39;−2.96]	−3.11 ^c^	[−4.70;−1.53]	-	-	−2.40 ^b^	[−3.96;-.84]	−1.29	[−3.00;0.42]
Secondary	-	-	−0.81	[−2.20;−0.56]	−0.28	[−1.74;1.18]	-	-	−0.76	[−2.23;.70]	−0.07	[−1.63;1.48]
Post-secondary	-	-	Ref.		Ref.		-	-	Ref.		Ref.	
Occupation												
Manual	-	-	-	-	−1.79^a^	[−3.40;−0.18]	-	-	-	-	−2.25 ^b^	[−3.89;−0.61]
Non manual	-	-	-		Ref.		-	-	-	-	Ref.	

^a^*p* < 0.05; ^b^
*p* < 0.01; ^c^
*p* < 0.001; Models adjusted by age, number of chronic diseases, self-rated health and depressive symptoms; CI = Confidence interval

**Table 4 ijerph-20-02789-t004:** Multivariate regression models assessing the influence of socioeconomic variables on the limitation subscale separated by gender.

	Men (n = 641)						Women (n = 721)					
	Model 1		Model 2		Model 3		Model 1		Model 2		Model 3	
	β	95% CI	β	95% CI	β	95% CI	β	95% CI	β	95% CI	β	95% CI
(Intercept)	76.77	[61.96;91.58]	76.87	[61.87;91.87]	77.03	[62.01;92.05]	94.36	[77.60;111.11]	94.78	[77.94;111.61]	96.59	[79.55;113.63]
Study site												
Natal	4.69 ^c^	[2.39,6.98]	4.57 ^c^	[2.25;6.90]	4.28 ^c^	[1.74;6.81]	5.17 ^c^	[2.70;7.63]	5.33 ^c^	[2.80;7.86]	4.94 ^c^	[2.12;7.77]
Manizales	6.69 ^c^	[4.47;8.91]	6.59 ^c^	[4.34;8.84]	6.35 ^c^	[3.96;8.74]	8.21 ^c^	[5.85;10.57]	8.39 ^c^	[5.96;10.83]	8.12 ^c^	[5.38;10.86]
St. Hyacinthe	3.45 ^c^	[1.69;5.21]	3.43 ^c^	[1.65;5.21]	3.29 ^c^	[1.45;5.13]	2.71 ^b^	[.85;4.56]	2.74 ^b^	[.87;4.60]	2.52 ^a^	[.56;4.47]
Kingston	Ref.		Ref.		Ref.		Ref.		Ref.		Ref.	
Income sufficiency												
Not/not very well	−2.44 ^a^	[−4.65;−0.24]	−2.30 ^a^	[−4.58;−0.03]	−2.39 ^a^	[−4.68;-.10]	−2.95 ^a^	[−5.22;−0.67]	−3.12 ^b^	[−5.47;−0.77]	−3.39 ^b^	[−5.77;−1.02]
Suitably	−1.42	[−3.13;.28]	−1.35	[−3.09;0.38]	−1.40	[−3.14;0.34]	-.76	[−2.57;1.05]	−0.86	[−2.71;0.98]	−0.99	[−2.86;0.88]
Very well	Ref.		Ref.		Ref.		Ref.		Ref.		Ref.	
Education level												
Less than secondary	-	-	−0.36	[−1.85;1.12]	−0.63	[−2.38;1.12]	-	-	0.60	[−1.21;2.41]	0.49	[−1.51;2.49]
Secondary	-	-	−0.38	[−1.90;1.14]	−0.53	[−2.15;1.08]	-	-	0.37	[−1.33;2.08]	0.25	[−1.57;2.07]
Post-secondary			Ref.		Ref.				Ref.		Ref.	
Occupation												
Manual	-	-	-	-	.52	[−1.25;2.30]	-	-	-	-	0.64	[−1.27;2.56]
Non manual	-	-	-	-	Ref.		-	-	-	-	Ref.	

^a^*p* < 0.05; ^b^
*p* < 0.01; ^c^
*p* < 0.001; Models adjusted by age, number of chronic diseases, self-rated health and depressive symptoms; CI = Confidence interval

## Data Availability

The datasets generated during and/or analyzed during the current study are available from the corresponding author upon reasonable request.

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
