# Peer review of "Gender-Specific Associations between Late-Life Disability and Socioeconomic Status: Findings from the International Mobility and Aging Study (IMIAS)"

_ijerph, 2023, doi:10.3390/ijerph20042789_

Round 1
Reviewer 1 Report
Thank you for the opportunity for this article. The topic is interesting and important. My only doubt and request for clarification is whether, in the opinion of the authors, it was possible to combine groups from different countries, since the authors indicate and emphasise “..in five cities with different cultures, economies, physical environments, and healthcare systems..”.
Reviewer 2 Report
This is an interesting and clear article on an important subject. The cross-country set-up, with some countries that are not often included in such studies, makes it quite valuable.
I have some questions, one of which is the reason for my recommendation of 'major revision'.
1) I find it surprising that the 'frequency subscale' is part of the LLFDL measure. It indicates whether persons engage or do not engage in certain activities, but not why. The reasons why they do not, may have nothing to do with disabilities or personal limitations. This in fact recognized in the 2nd paragraph of the Discussion (lines 261-76). As the LLFDL is a recognized and validated scale, I do not criticize this choice, but some words to motivate it might be useful. In Europe, where I work, it is more common to use limitation in ADLs and IADLs as measures of disability. (Activities of Daily Living and Instrumental Activities of Daily Living)
2. This is my most critical comment: I do not understand why 'Analytical models were adjusted for age, number of chronic diseases, self-rated health and depression symptoms.' In my view, these variables (except age) are intermediate between socio-economic status and limitation: e.g., because people had arduous jobs, they have more chronic conditions, and therefore are limited in activities. No reference or argument is given for this choice. The inclusion of these covariates may mean that the estimated effects underestimate the true effect of the SES on limitations. I would therefore recommend that models are estimated without those covariates. The models with those covariates can serve to show to what extent the effects of SES are mediated by those covariates.
3. Implicit in the models is the assumption that all variables have the same effect in all Study sites. This might be questioned, given the large differences in the social and economic circumstances between those sites. While the number of observations does not allow (I guess) to estimate models for each site, some sensitivity tests are required.
4) The introduction mentions Tirana (Albania) as a study site, but in the results Tirana is absent. Why?
Round 2
Reviewer 2 Report
Thanks for your polite response, though you do not really address the points I raised (except the last one about Tirana). In particular, in reaction to my question "why analytical models were adjusted for age, number of chronic diseases, self-rated health and depression symptoms", you answer:
"To assess this possibility longitudinal data and performance of a mediation analysis is needed which is beyond the scope of this work mostly because of data limitation."
In itself true, but not really to the point. The question is whether the (total) effect of SES is best (most validly) estimated with or without those controls. I argue that if those variables (e.g. chronic diseases) are mediating variables, it would be best without. You do not really answer that point.
I regard your choice as unfortunate rather than wrong. Further discussion seems pointless, and your study is worthwhile. Therefore I recommend acceptance.
